# The effects of a yoga intervention on balance and flexibility in female college students during COVID-19: A randomized controlled trial

Xue Luo[1,2]*, Xu Huang[1]

**1** Department of Business Administration, Southwestern University of Finance and Economics, Chengdu, Sichuan, China, **2** Department of Physical Education, Chengdu Normal University, Chengdu, Sichuan, China

* cdtyxylx@163.com

## Abstract

Female college students are more likely to have a strong stress response to the COVID-19 pandemic, which seriously affects their health and merits greater attention. The present study is a randomized controlled trial carried out during the COVID-19 pandemic. The purpose of this study was to explore whether a yoga intervention could have a positive impact on balance and flexibility and be the primary form of home exercise for female college students in China. Forty female college students were randomly selected for the study. After 16 weeks of yoga intervention, the ability of the yoga group to balance on one leg improved by 5.35 seconds (using the single-limb stance test) and 5.7 seconds (using the Romberg test). The flexibility of the lower back and legs increased by 3.5 cm (using the sit and reach test), and the flexibility of the groin and hip increased by 6 cm (using the Splits test). The balance ability and flexibility of the control group before and after the experiment were not significantly different (p>0.05). These results suggest that during the COVID-19 outbreak when access to public facilities was limited, a yoga intervention could have a positive impact on health and be the main form of home exercise for female college students. Schools and government departments could design online yoga classes to encourage individuals to actively participate in yoga to stay healthy.

## Introduction

COVID-19 is a major public health crisis with the fastest spread, the widest infection, and the greatest difficulty in prevention and control since the founding of New China; therefore, it is a major test for China [1]. The safety and health of human life are under major threat [2, 3]. Some studies have shown that college students are more likely to have strong stress reactions than ordinary employees in the face of the COVID-19 pandemic and related measures, such as online learning and remote employment [4]. Compared with male college students, female college students have been more susceptible to the external environment during the pandemic [5, 6] and are more likely to rely on others to help them [4, 7]. This, together with the more

**Data Availability Statement:** The data that support the findings of this study are within the Supporting Information files.

**Funding:** This research was funded by the Ministry of Education in China Project of Humanities and Social Sciences (Grant number 21YJC890020), Chengdu Normal University Teaching Reform Project (Grant number 2021JG49), Chengdu Normal University Model Course "Curriculum Civics" (Grant number XJKCSZKC2023), Chengdu Normal University Project (Grant number CS21SC03), Chengdu World Event City Research Center (Grant number CDMC2022B05), Sichuan Province first-class undergraduate online course "Teach you to practice good temperament - Yoga" (Grant number 2022SJYLKC02), all awarded to XL.

**Competing interests:** The authors have declared that no competing interests exist.

severe employment pressure faced by female college students [8], has resulted in a greater health threat to female college students [9]. During such public health crises, we need to pay more attention to female college students and implement health interventions to help them through difficult times.

Balance and flexibility are important components of health [10, 11]. Balance is defined as the postural control that provides a stable support base when remaining at rest or in motion [12]. Balance disorders affect independence and mobility in activities of daily living and increase the likelihood of falls during changes in body position [13]. Yoga is an umbrella term for physical, mental, and spiritual disciplines whose practices include an emphasis on breathing, meditation, and postures that positively affect balance and flexibility. Balance was measured using the single-limb stance test [14–16] and the Romberg test [12, 17].

Flexibility is defined as the ability to achieve certain body mechanics or move body parts through a range of motion [18, 19]. Increased flexibility is one of the fastest attainable benefits of regular yoga practice, as it is based on the progressive extension of connective tissue and muscles around joints and bones by statically maintaining yoga asanas within the existing range of motion [18]. In most studies, the sit and reach test is selected as the measurement index of flexibility [20, 21]. In this study, the sit and reach test was used to evaluate the flexibility of the waist and legs, and the Splits test [22] was used to evaluate the flexibility of the groin and hip.

In recent years, yoga has become a widely popular exercise among women, and several studies have reported the benefits of yoga for balance and flexibility in women [23, 24]. However, those aforementioned studies have several limitations. First, most of the studies focused on the impact of yoga on the health of women in the general population and did not target the key population affected by the COVID-19 pandemic, which is Chinese female college students aged 17 to 24. There are scant studies on whether yoga was a timely and effective intervention in the health of female college students during the COVID-19 outbreak. Second, most studies chose the sit and reach test as the index to measure the flexibility of participants. In addition to the sit and reach test index, the present study also added the Splits test index to measure the flexibility of the participants' groin and hip, expanding the range of flexibility measurement index selection to more comprehensively measure the flexibility of participants.

This study investigated the effects of yoga intervention on balance and flexibility in Chinese female university students aged 17–24 years, a key population affected by the COVID-19 pandemic, to address a significant gap in the literature. This study sought to answer the following specific questions: can yoga positively contribute to the balance and flexibility of female college students during the COVID-19 pandemic? To what extent does yoga practice affect the balance and flexibility of female college students? Can yoga be the main mode of home exercise for female college students?

The novel contributions of this study are as follows: first, this study is a randomized controlled trial conducted during the COVID-19 pandemic. COVID-19 poses a serious health threat to female college students, and our study of the effects of a yoga intervention on balance and flexibility in female college students has important theoretical and practical implications. Our study fills a gap in the implementation of yoga interventions for female university students in China during the COVID-19 outbreak, and our findings could help them stay healthy during similar public health crises. Second, most studies use the sit and reach test index to measure participant flexibility. In addition to the sit and reach test index, this study also adds the Splits test index to measure the flexibility of the participants' groins and hips, expanding the selection range of flexibility measurements to more comprehensively assess participants' flexibility.

Based on the above analysis, the following hypotheses are proposed in this study.

- Hypothesis 1 (H1): the yoga intervention will positively contribute to the balance of female college students during the COVID-19 pandemic.

- Hypothesis 2 (H2): the yoga intervention will have a positive effect on the flexibility of female college students, and yoga can be used as the main form of home exercise for female college students during the COVID-19 pandemic.

## Materials and methods

### Participants

Participants were recruited via flyers; for a detailed description of the flyers, please see Document S1. This study was approved by the Ethics Committee of the College of Physical Education, Chengdu Normal University (approval number: YJJKGY [2020] 07).

Inclusion criteria are as follows: female university students enrolled in school, who have not practiced yoga before, will not perform other exercises during the experiment; are serious about attending yoga class on time, are not practicing specific balance or flexibility exercises, and volunteer to participate in this study and sign an informed consent form.

Exclusion criteria are as follows: chronic illness, physical disability, current or previous yoga experience, physical condition unsuitable for yoga, physical condition unsuitable for exercise training, hypertension, hypoglycemia, cerebellar disease, joint disease, or eye disease [25].

### Intervention

The yoga group practiced under the guidance of a yoga instructor for 70 minutes twice a week for 16 weeks. The yoga sessions in this study were designed according to a yoga textbook and were validated by a panel of yoga instruction experts. The purpose of the validation was to ensure that the selected yoga practice content was appropriate and safe for female college students who were severely affected by COVID-19. Each session included 10 minutes of breathing and centering techniques, 10 minutes of warm-up exercises, 40 minutes of posture practice, and 10 minutes of resting techniques (Table 1).

The control group watched 10 minutes of yoga videos each week. These yoga videos included Hatha, Aerial, Ashtanga, Iyengar, Flow, Yin and Dance yoga. The control group watched yoga videos for 16 weeks.

### Measures

In this study, a panel composed of four university yoga teachers and four yoga instructors was formed. The expert group reached a consensus on the measurement indicators of balance ability and flexibility (Table 2). Within one week of the yoga intervention, both the yoga and control groups were asked to take the balance and flexibility test again to compare the pretest and posttest scores.

**Balance measurements.** The single-limb stance test and the Romberg test were used to assess the balance ability of the participants enrolled.

Single-limb stance test: the measurement method for the single-limb stance is to stand naturally with eyes closed and lift any foot after hearing the "start" command, at which time the tester starts timing. When the participant's supporting foot moves or the lifted foot touches the ground, the tester stops timing. The time is recorded in seconds, the test is performed three times, and the highest value is used as the final result. The longer the time is, the better the balance ability.

**Table 1. Contents of yoga practice for the 16-week training program.**

|  | Position | Yoga Poses | Time |
|---|---|---|---|
| I | Breathing and centering techniques | Pranayama (Breathing and Meditation) | 10 min |
|  | Warm-up exercises | Joint movement exercises | 10 min |
| II Postures (Asanas) | Standing postures | Utkatasana (Chair Pose)<br>Uttanasana (Standing forward bend)<br>Vrksasana (Tree Pose)<br>Garudasana (Eagle Pose)<br>Naṭarajasana (Dancer Pose)<br>Virabhadrasana I (Warrior Pose)<br>Virabhadrasana (Warrior II)<br>Virabhadrasana III (Warrior III)<br>Utthita Trikonasana (Triangle Pose)<br>Parivrtta Baddha Parsvakonasana (Revolved Side Angle Pose) | 40 min |
|  | Sitting postures | Dasana (Staff Pose)<br>Janu Sirsasana (Head-to-Knee Pose)<br>Parivrtta Janu Sirsasana (Revolved Head-to-Knee Pose)<br>Baddha Konasana (Bound angle Pose)<br>Gomukhasana (Cockface Pose)<br>Navasana (Boat Pose)<br>Ardha Matsyendrasana (Half Lord of the Fishes Pose) |  |
|  | Kneeling Postures | Balasana (Child's Pose)<br>Supta Virasana (Reclining Hero or Heroine Pose)<br>Ustrasana (Camel Pose)<br>Eka Pada Rajakapotasana (One-Legged King Pigeon Pose)<br>Parighasana (Gate Pose)<br>Simha asana (Lion pose) |  |
|  | Prone postures | Savasana (Corpse Pose)<br>Apanasana (Knees-to-Chest)<br>Setu Bandha Sarvangasana (Bridge Pose)<br>Setubandha Sarvangasana (Shoulder stand)<br>Viparita Karani (Legs-Up-the-Wall Pose)<br>Halasana (Plow Pose)<br>Jathara Parivrtti Asana (Belly Twist) |  |
|  | Supine postures | Bhujangasana (Cow Pose)<br>Dhanurasana (Low Pose)<br>Salabhasana (Locust Pose) |  |
| III | Deep relaxation technique | Rest Technique | 10 min |

Romberg test: Romberg's measurement method is to stand with bare feet and eyes closed. After hearing the "start" command, lift either foot and put it on the heel of the other foot, without the toes touching the ground, with the arms hanging down naturally. At that time, the tester begins timing. When either of the participant's two feet moved or the body became unstable, the tester stopped timing. The time is recorded in seconds, the test is performed three times, and the highest value is used as the final result. The longer the time is, the better the balance ability.

**Table 2. Measures of balance and flexibility.**

| Measurement Items | Measurement Indicators | Units of Measurement |
|---|---|---|
| Balance measurements | Single-limb stance test | seconds |
|  | Romberg test | seconds |
| Flexibility measurements | Sit and reach test | cm |
|  | Splits test | cm |

**Flexibility measurements.** The sit and reach test and the Splits test were used to evaluate the flexibility of the participants.

Sit and reach test: the participants' lumbar and leg flexibility were measured by the sit and reach test. The participant sits on a yoga mat in the test area, both legs straight, two feet flat against the heavy plate, upper body forward bending, arms straight, with the middle finger of both hands gradually pushing the cursor (without any sudden pushes) until they cannot push any further. The tester checks the longitudinal plate plane for the zero point, the longitudinal plate inward for the negative value, and the longitudinal plate forward for the positive value. The measurement is recorded in centimeters, the test is performed three times, and the highest value is used as the final result. The greater the value is, the better the flexibility.

Splits test: the participants' groin and hip flexibility were measured by the Splits test. The measurement method for the Splits test has the participant straighten both legs and Splits one to the front and the other back in a straight line, with the hips and both legs on the ground. The distance between the groin and the ground is measured in centimeters. The test is performed three times, and the maximum value is taken as the final result. The smaller the distance between the groin and the ground, the better the flexibility.

## Statistical analysis

IBM SPSS Statistics Version 26.0 was used to analyze the data. The demographic characteristics of the participants were described using descriptive statistics. In this study, an analysis of variance (ANOVA) with a mixed design of 2 (group: yoga group, control group) × 2 (time: pretest, posttest) was used to analyze the effects of the yoga intervention on the balance ability and flexibility of female college students seriously affected by COVID-19 and to verify whether yoga can be used as a primary method of home exercise for female college students during the COVID-19 pandemic. A value of $p < 0.05$ was considered significant.

## Results

Fifty-seven participants who came to the yoga consultation were selected, and 40 participants were included in the study. We assumed a sample size ratio of 1:1 for the yoga and control groups, a test level of $\alpha = 0.05$ (two-sided), and a test efficacy of $1-\beta = 0.9$. Using SAS 9.4 software, the total sample size required for this study was calculated as a total of 40 participants (20 in the yoga group and 20 in the control group) (Fig 1). We paid participants who completed all tasks in this study 200 RMB, which is relatively high compared to other experiments of the same type in the Wenjiang area of Chengdu; all participants completed the study. Forty participants were randomly divided into the yoga and control groups. Before each yoga practice, the 40 participants were required to present a negative nucleic acid report certificate. The study was conducted from April 2020 to July 2020.

Participants (n = 40) were between the ages of 17 and 24 years, had not practiced yoga before and were college students from the Southwest University of Finance and Economics, Chengdu Normal University, Sichuan Agricultural University, and Chengdu University of Traditional Chinese Medicine. The characteristics of the participants are shown in Table 3. For the overall sample, the mean age, height, and weight were 20.35 years (standard deviation [SD], 2.37 years), 161.15 cm (SD, 2.99 cm), and 52.05 cm (SD, 3.21 cm), respectively. There were no significant differences between the two groups in terms of grade, school, age, height, or weight (Table 3; P > 0.05).

To confirm the first hypothesis (H1) that the yoga intervention could positively contribute to the balance ability of female college students during the COVID-19 outbreak, the following test results were compared. As shown in Table 4 and Fig 2, the repeated measures variance

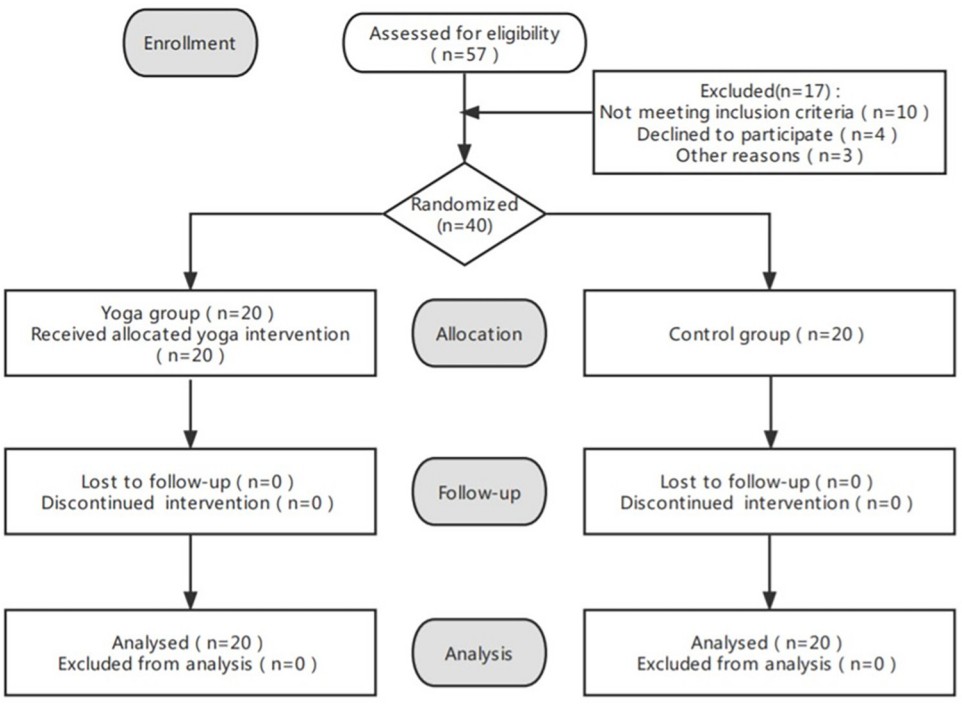

**Fig 1. Recruitment flow chart.**

revealed a significant interaction between group and time of the single-limb stance test (F = 122.25, p < 0.05), with Cohen's d of 2.59 for the pretest and posttest of the single-limb stance in the yoga group, and the effect size was 0.79, p < 0.05. Cohen's d for the control group single-limb stance pretest and posttest was -0.14, and the effect size was -0.07, p > 0.05. Table 4 and Fig 3 show that there was a significant interaction between group and time for Romberg's test (F = 133.91, p < 0.05), with Cohen's d for Romberg pretest and posttest in the yoga group being 2.54 and effect size being 0.79, p < 0.05. Cohen's d for the Romberg pretest and posttest in the control group was 0, and the effect size was 0, p > 0.05. The results suggest that yoga intervention can positively contribute to the balance ability of female college

**Table 3. Characteristics of the participants (n = 40).**

| Variables | Minimum | Maximum | Yoga Group (n = 20) Mean (±SD) | Control Group (n = 20) Mean (±SD) | F value | P value | T value |
|---|---|---|---|---|---|---|---|
| **Grade**<br>Freshman<br>Sophomore<br>Junior<br>Senior | | | 5 (12.5%)<br>7 (17.5%)<br>3 (7.5%)<br>5 (12.5%) | 5 (12.5%)<br>6 (15.0%)<br>4 (10%)<br>5 (12.5%) | .008 | .929 | -.138 |
| **School**<br>Southwestern University of Finance and Economics<br>Chengdu Normal University<br>Sichuan Agricultural University<br>Chengdu University of TCM | | | 5 (12.5%)<br>7 (17.5%)<br>3 (7.5%)<br>5 (12.5%) | 5 (12.5%)<br>6 (15.0%)<br>4 (10%)<br>5 (12.5%) | .008 | .929 | -.138 |
| **Age (years)** | 17 | 24 | 20.30±2.34 | 20.40±2.46 | .033 | .858 | -.132 |
| **Height (cm)** | 155.65 | 166.01 | 161.31±2.77 | 161.00±3.27 | .724 | .400 | .314 |
| **Weight (kg)** | 45.76 | 58.02 | 52.90±3.06 | 51.19±3.20 | .359 | .553 | 1.731 |

**Table 4. Comparison of balance and flexibility between the yoga and control groups.**

| Variables | Measurement Indicators | Yoga Group | | Control Group | |
|---|---|---|---|---|---|
| | | Pretest Mean (±SD) | Posttest Mean (±SD) | Pretest Mean (±SD) | Posttest Mean (±SD) |
| Balance | Single-limb stance (s) | 12.65±1.46 | 18.00±2.53 | 13.40±1.60 | 13.20±1.20 |
| | Romberg (s) | 10.40±1.47 | 16.10±2.81 | 11.50±1.64 | 11.50±1.19 |
| Flexibility | Sit and reach (cm) | 0.30±0.46 | 3.80±1.72 | 0.43±0.44 | 0.45±0.42 |
| | Splits (cm) | 11.25±2.07 | 5.25±1.55 | 11.50±1.99 | 11.00±1.59 |

students. After 16 weeks of yoga intervention, female college students' balance improved by 5.35 s (single-limb stance test) and 5.70 s (Romberg test).

To test the second hypothesis (H2) that the yoga intervention would have a positive impact on female college students' flexibility and that yoga could be the primary form of home exercise for female college students during the COVID-19 pandemic, the following test results were considered. As shown in Table 4 and Fig 4, the repeated measures variance revealed a significant interaction between group and time of sit and reach test (F = 139.51, $p < 0.05$), with Cohen's d of 2.78 for the pretest and posttest of sit and reach in the yoga group and effect size of 0.81, $p < 0.05$. Cohen's d for the control group sit and reach pretest and posttest was 0.05, and the effect size was 0.02, $p > 0.05$. Table 4 and Fig 5 show that there was a significant interaction between group and time for the Splits test (F = 294.74, $p < 0.05$), with the yoga group Cohen's d for the Splits pretest and posttest being -3.28 and the effect size being -0.85, $p < 0.05$. Cohen's d for Splits pretest and posttest in the control group was -0.28, and the effect size was -0.14, $p > 0.05$. The results suggest that yoga can be used as the main form of home exercise for female college students during the COVID-19 pandemic and that yoga practice can have a positive impact on the flexibility of female college students. After 16 weeks of yoga intervention, female college students' flexibility improved by 3.50 cm (Sit and reach test) and 6 cm (Splits test).

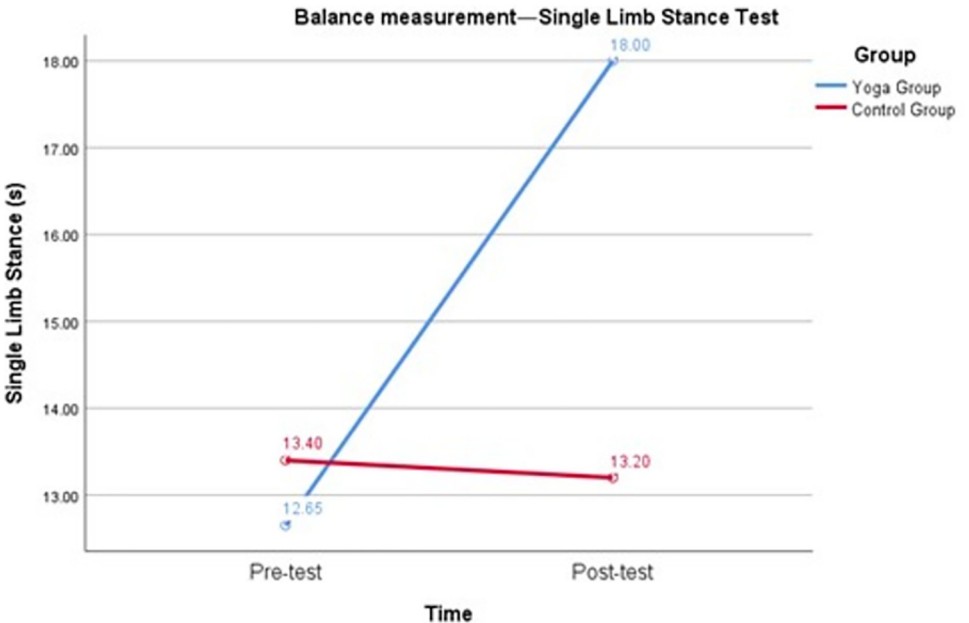

**Fig 2. Balance measurement—Single-limb stance test.**

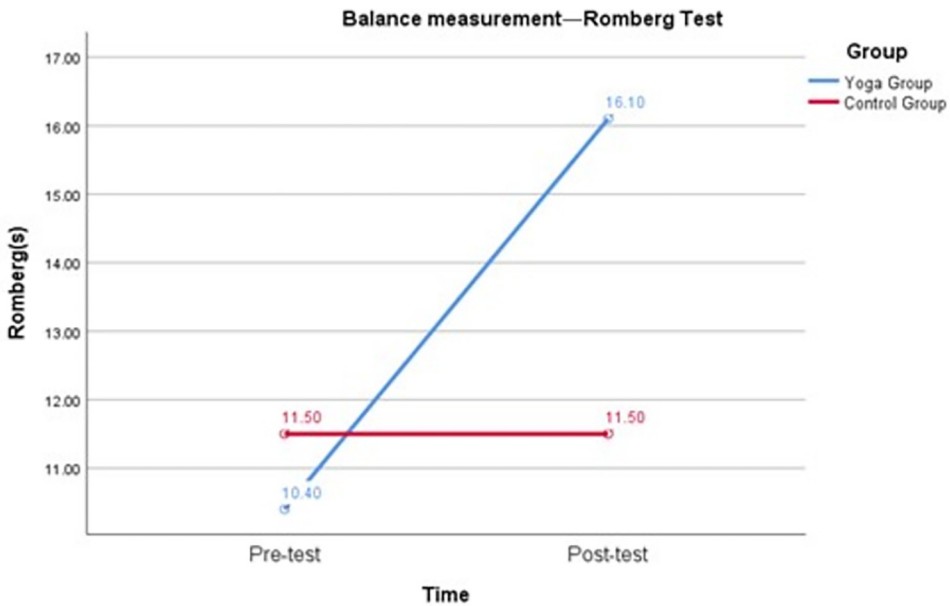

**Fig 3. Balance measurement—Romberg test.**

## Discussion

COVID-19 poses a greater health threat to female college students than male students. How to help female college students stay healthy during the COVID-19 pandemic is a question worthy of study. Our study evaluated the effects of a 16-week yoga intervention on the balance and flexibility of female college students using the single-limb stance test, Romberg test, sit and reach test, and Splits test. We found that the yoga intervention positively contributed to the

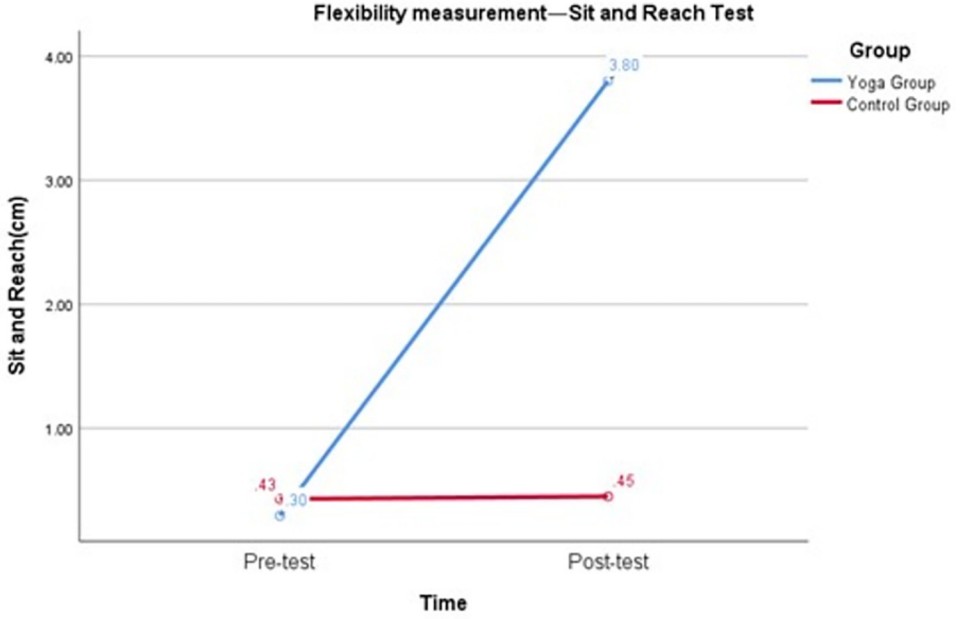

**Fig 4. Flexibility measurement—Sit and reach test.**

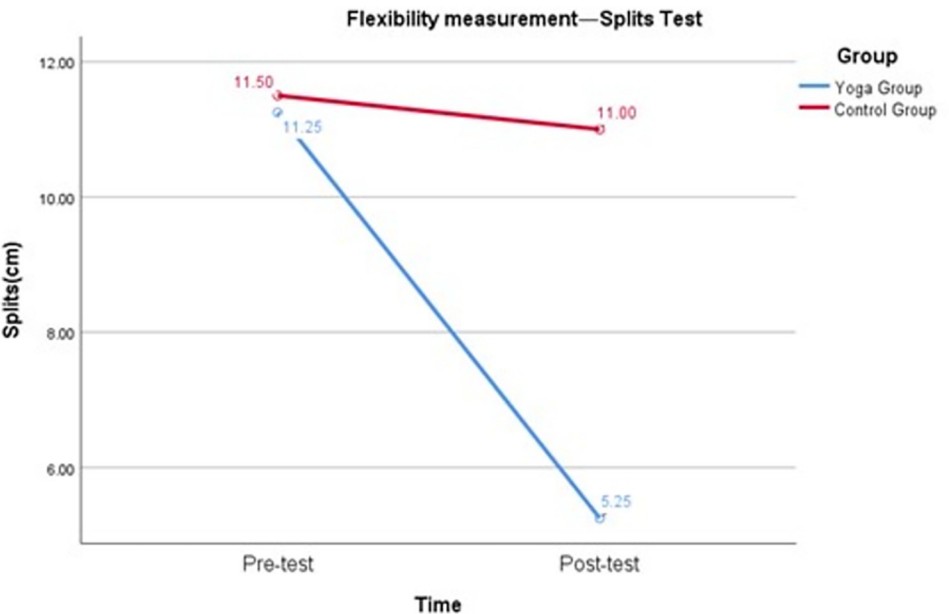

**Fig 5. Flexibility measurement—Splits test.**

balance of female college students during the COVID-19 pandemic, confirming H1. In addition, we found that yoga practice could have a positive impact on the flexibility of female university students, and yoga could be the main form of home exercise for female university students, confirming H2.

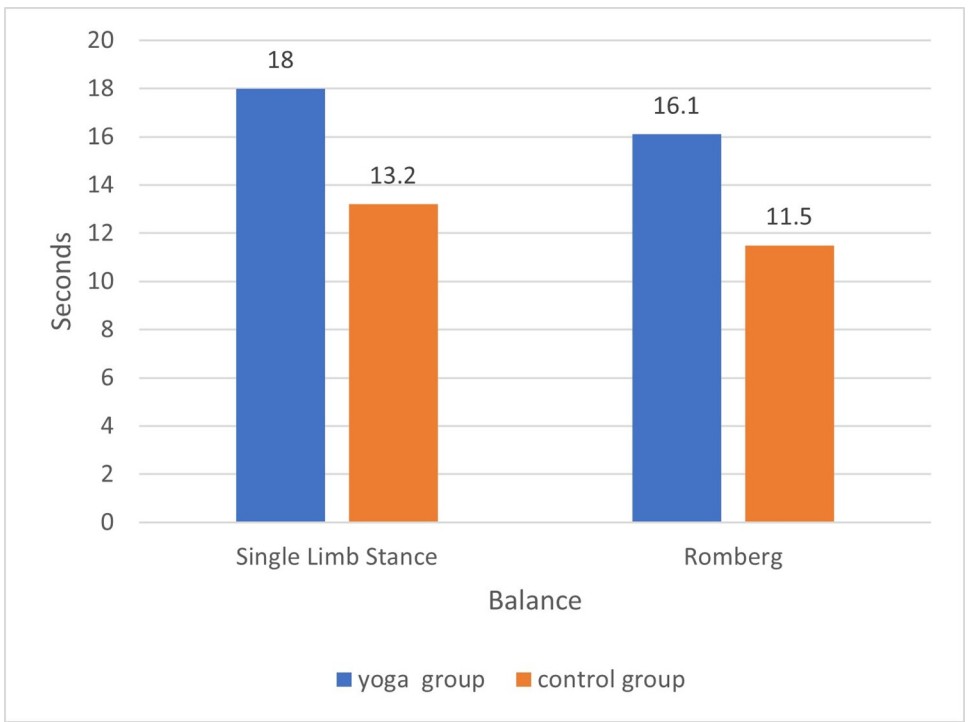

**Fig 6. Posttest comparison of balance.**

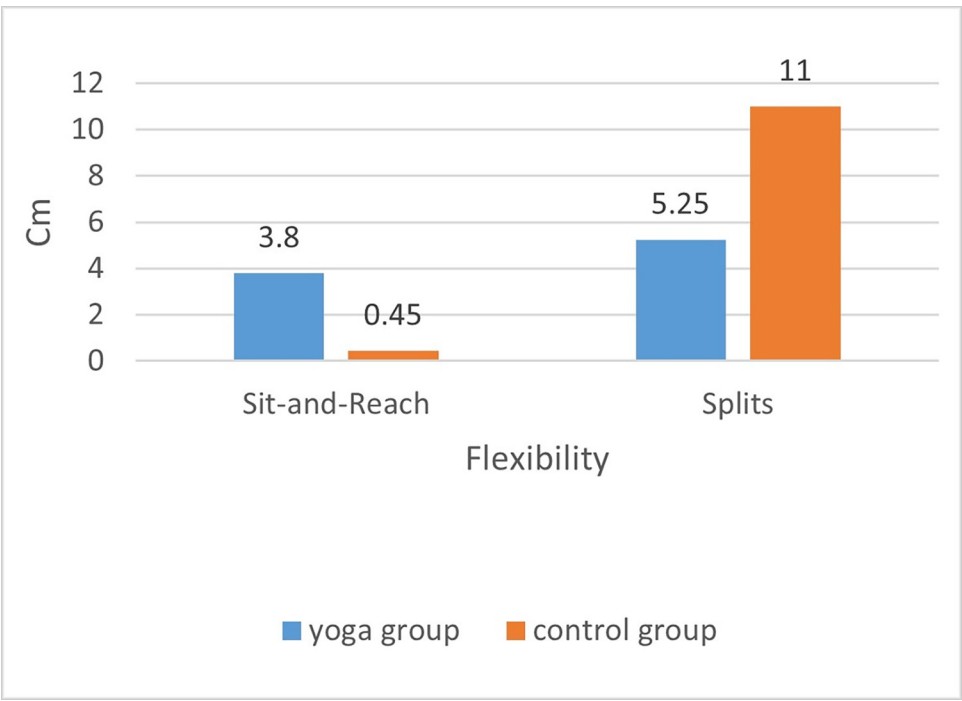

**Fig 7. Posttest comparison of flexibility.** * The smaller the value of the Splits was, the better the flexibility of the groin, hip, and leg.

The first result showed that the balance of female college students improved by 5.35 s (single-limb stance test) and 5.70 s (Romberg test) after 16 weeks of yoga intervention during the pandemic (Fig 6). Although it has been suggested that practicing yoga can improve balance in women [12, 13], our findings differ from those of other studies in that this study was a yoga intervention conducted during the COVID-19 pandemic. Our study fills a gap in the lack of yoga interventions implemented with Chinese female college students during the COVID-19 pandemic. The results of this study suggest that the implementation of a yoga intervention with female college students during the pandemic significantly improved their balance and had a positive impact on their health.

In this study, we chose the single-limb stance Test and the Romberg Test as indicators of balance ability because they both appear to be good predictors of advanced balance [16]. Both metrics require participants to use their core to maintain balance during balance measurements while the scapula remains stable. Prior to the yoga intervention, participants had tension in their shoulders and neck and were unable to maintain stability in their upper and lower extremities. As a result of yoga practice, participants' gluteus maximus, quadriceps and calf muscle strength were effectively developed, lower extremity muscle strength was enhanced and balance was improved.

The second result showed that the flexibility of female college students increased by 3.50 cm (sit and reach test) and 6 cm (Splits test) after 16 weeks of yoga intervention during the pandemic (Fig 7). Some studies have suggested that practicing yoga can improve flexibility in women; for example, after practicing yoga for 8 weeks, participants improved their lower body flexibility by 4.1 cm [26], 0.9 cm [27], and 10.0 cm [28]. In these studies, when measuring flexibility, the primary flexibility indices were the sit and reach test. Our study differs from these studies in that in addition to the sit and reach test index, our study added the Splits test index

to measure flexibility in the groin and hips of the participants. Another difference between our study and previous studies is that our study is a randomized controlled trial conducted during the COVID-19 pandemic and focused on female college students. Our findings make a significant theoretical and practical contribution to the implementation of timely and effective health interventions for female college students during the COVID-19 pandemic, when yoga could be used as a primary form of home exercise.

Yoga poses include many stretching exercises. The participant's body is slowly stretched while holding a certain pose [26]. The muscles and ligaments are gradually stretched, and the participant's body is fully stretched [27]. Proper stretching can help female university students move their joints and muscles better. After 16 weeks of yoga practice, the flexibility of the participants' bodies improved.

## Conclusion

In this study, we assessed how much a 16-week yoga intervention could affect the balance and flexibility of female college students during the COVID-19 pandemic using the single-limb stance test, Romberg test, sit and reach test, and Splits test. We found that the yoga intervention could positively affect the balance and flexibility of female college students during the COVID-19 pandemic and that yoga could be the primary form of home exercise for female college students.

According to the findings of this study, schools and government departments should give more care and support to female college students during the COVID-19 pandemic and future public health crises by designing online yoga classes and activities to encourage them to actively participate in yoga practice to stay healthy.

## Supporting information

**S1 File. Flyers.**
(DOC)

**S2 File. Data file.**
(XLSX)

## Author Contributions

**Conceptualization:** Xue Luo.

**Data curation:** Xue Luo.

**Funding acquisition:** Xue Luo.

**Investigation:** Xu Huang.

**Methodology:** Xue Luo, Xu Huang.

**Supervision:** Xu Huang.

**Writing – original draft:** Xue Luo.

**Writing – review & editing:** Xu Huang.

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
