## [Decision Letter · Decision Letter 0]

8 Jan 2023

PONE-D-22-33655The effects of a yoga intervention on balance and flexibility in female college students during COVID-19: A randomized controlled trialPLOS ONE

Dear Dr. Luo,

Thank you for submitting your manuscript to PLOS ONE. After careful consideration, we feel that it has merit but does not fully meet PLOS ONE’s publication criteria as it currently stands. Therefore, we invite you to submit a revised version of the manuscript that addresses the points raised during the review process.

The manuscript should be revised according to the reviewers' suggestions.

We look forward to receiving your revised manuscript.

Kind regards,

Alessandro de Sire, M.D.

Academic Editor

PLOS ONE

Additional Editor Comments:

The manuscript should be revised according to the reviewers' suggestions.

Reviewers' comments:

Reviewer's Responses to Questions

**Comments to the Author**

1. Is the manuscript technically sound, and do the data support the conclusions?

Reviewer #1: Partly

Reviewer #2: Yes

2. Has the statistical analysis been performed appropriately and rigorously? 

Reviewer #1: Yes

Reviewer #2: Yes

3. Have the authors made all data underlying the findings in their manuscript fully available?

Reviewer #1: Yes

Reviewer #2: Yes

4. Is the manuscript presented in an intelligible fashion and written in standard English?

Reviewer #1: Yes

Reviewer #2: Yes

5. Review Comments to the Author

Reviewer #1: Dear Sirs,

Your paper seems interesting since it deals with a social and health emergency, that is the impact of Covid-19 pandemic on female college students’ balance and flexibility, and it also offers a possible solution, that is yoga.

Nevertheless, some concerns have to be addressed.

ABSTRACT

It is clear and concise.

INTRODUCTION

This section seems repetitive and pleonastic. Lines 117-120 are needless, and the initial hypotheses of the authors are missing.

METHODS

Please specify the recruitment method. Which flyer did you use? What information were included in it?

How do you explain the absence of drop outs?

How did you perform the sample size calculation? Was the sample a convenience one? Please specify it.

Line 164 and everywhere inside the text: no capital letters after the colon

Lines 194-195: please reformulate the sentence.

It is not well expressed what treatment underwent the control group. This is mandatory in a randomized controlled trial. Please clearly explain it.

DISCUSSION AND CONCLUSIONS

You should revise the structure of this sections. Conclusions are too long, and a substantial part of them could be moved to the discussion which seems poor and needs to be better contextualised. Moreover, is not clear if You aimed at improving psychological health, which largely mentioned in the introduction, or at improving the balance and flexibility tout court. To better describe the context in which you did this research, highlighting the peculiarities of Covid-19 pandemic and female teenagers psychophysical profile, I suggest the following references:

Farì, G., Di Paolo, S., Ungaro, D., Luperto, G., Farì, E., & Latino, F. (2021). The impact of covid-19 on sport and daily activities in an italian cohort of football school children. International Journal of Athletic Therapy and Training, 26(5), 274-278. doi:10.1123/ijatt.2020-0066

Notarnicola, A., Farì, G., Maccagnano, G., Riondino, A., Covelli, I., Bianchi, F. P., . . . Moretti, B. (2019). Teenagers’ perceptions of their scoliotic curves. an observational study of comparison between sports people and non- sports people. Muscles, Ligaments and Tendons Journal, 9(2), 225-235. doi:10.32098/mltj.02.2019.11

de Sire, A., Demeco, A., Marotta, N., Spanò, R., Curci, C., Farì, G., . . . Ammendolia, A. (2022). Neuromuscular impairment of knee stabilizer muscles in a COVID-19 cluster of female volleyball players: Which role for rehabilitation in the post-COVID-19 return-to-play? Applied Sciences (Switzerland), 12(2) doi:10.3390/app12020557

REFERENCES

Please check that references are all represented according to editorial guidelines.

Best regards

Reviewer #2: The article is of scientific interest and in line with the purposes of the journal. The authors guidelines have been met and the manuscript does not require proofreading in English by a native speaker. There are some concerns to fix.

KEYWORDS

In order to increase the visibility of the article, do not use keywords already present in the title.

INTRODUCTION

The backround is well described.

The purpose of the study is clearly stated.

"h are facing major threats.". You need a reference. I suggest inserting the following reference:

- de Sire A, Marotta N, Agostini F, Drago Ferrante V, Demeco A, Ferrillo M, Inzitari MT, Pellegrino R, Russo I, Ozyemisci Taskiran O, Bernetti A, Ammendolia A. A Telerehabilitation Approach to Chronic Facial Paralysis in the COVID -19 Pandemic Scenario: What Role for Electromyography Assessment? J Pers Med. 2022 Mar 19;12(3):497 . doi:10.3390/jpm12030497. PMID: 35330496; PMC ID: PMC8949994

METHODS

The inclusion and exclusion criteria have been adequately described.

The measurements are in accordance with the aim of the study.

Statistical analysis was adequately described and performed.

"57 participants who came to the yoga consultation

124 were selected, and 40 participants were enrolled in the study....of 20 women after screening (Figure 1)......articipants (n = 40). The sociodemograph...(Table 1; p > 0.05). ". Move this information, and also figure 1, to the results section. In the methods section, there must be no references to the number of patients, but only inclusion and exclusion criteria.

RESULTS AND DISCUSSION

The results are written very fluently for the reader and adequately argued in the discussion section.

CONCLUSIONS

Appropriate.

REFERENCES

References are recent and relevant to the subject. Add the suggested reference.

TABLES

The tables are clear and adequately complement the text.

FIGURES

The figures are of good quality.

6. PLOS authors have the option to publish the peer review history of their article (what does this mean?). If published, this will include your full peer review and any attached files.

Reviewer #1: No

Reviewer #2: No

---

## [Author Response · Author response to Decision Letter 0]

7 Feb 2023

Dear Alessandro de Sire, M.D. Academic Editor, Reviewers 1 and 2:

Thank you for your comments and suggestions on our manuscript. The comments and suggestions are all valuable and very helpful for revising and improving our study, as well as guiding our research. We have studied the comments carefully and have revised this manuscript. Thank you very much for giving us this valuable opportunity. We have responded to each of your questions below and changes to the revised manuscript appear in red.

Alessandro de Sire, M.D. Academic Editor:

1. Question: Please ensure that your manuscript meets PLOS ONE's style requirements, including those for file naming. 

1. Answer: Thank you for your suggestions. We have revised the manuscript in accordance with the formatting requirements of PLOS ONE.

2. Question: We note that the grant information you provided in the ‘Funding Information’ and ‘Financial Disclosure’ sections do not match. When you resubmit, please ensure that you provide the correct grant numbers for the awards you received for your study in the ‘Funding Information’ section.

2. Answer: Thank you. ‘Funding Information’ and ‘Financial Disclosure’ has been changed, please refer to the financial disclosure and funding information in the cover letter. 

‘Funding Information’ and ‘Financial Disclosure’: This research was funded by Chengdu Normal University Teaching Reform Project, Grant number 2021JG49.Chengdu Normal University Model Course "Curriculum Civics”, Grant number XJKCSZKC2023.Chengdu Normal University Project, Grant number CS21SC03. Chengdu World Event City Research Center, Grant number CDMC2022B05. Sichuan Province first-class undergraduate online course "Teach you to practice good temperament - Yoga", Grant number 2022SJYLKC02.

3. Question: Your ethics statement should only appear in the Methods section of your manuscript. If your ethics statement is written in any section besides the Methods, please move it to the Methods section and delete it from any other section. Please ensure that your ethics statement is included in your manuscript, as the ethics statement entered into the online submission form will not be published alongside your manuscript.

3. Answer: Thank you. I have followed your request to have the ethics statement appear only in the methods section of the manuscript and have removed the ethics statement from the rest of the manuscript. 

Reviewer #1:

1. Question: INTRODUCTION

This section seems repetitive and pleonastic. Lines 117-120 are needless, and the initial hypotheses of the authors are missing.

1. Answer: Thank you for your valuable suggestions. We have reworked the introduction to remove duplications as you suggested. Lines 117-120 have been removed, and the research hypotheses have been added. Please review the introduction in red. Thank you.

2. Question: METHODS

Please specify the recruitment method. Which flyer did you use? What information were included in it? How do you explain the absence of drop outs? How did you perform the sample size calculation? Was the sample a convenience one? Please specify it. Line 164 and everywhere inside the text: no capital letters after the colon Lines 194-195: please reformulate the sentence. It is not well expressed what treatment underwent the control group. This is mandatory in a randomized controlled trial. Please clearly explain it. 

2. Answer: Thank you for the questions; they are good ones. I will now respond to each of the questions you have asked. Thank you.

Question: Please specify the recruitment method. Which flyer did you use? What information were included in it?

Answer: The flyer we use is included in the supporting information of the manuscript; please refer to the S1 document. The details of the flyer are as follows.

Recruiting Participants for the Yoga Intervention Health Study

[Experimental content]

Effects of yoga intervention on balance and flexibility of female college students during COVID-19.

[Experiment in brief]

The COVID-19 pandemic poses a serious health threat to female college students and deserves more attention. This study is a randomized controlled trial conducted during the COVID-19 pandemic. The purpose of this trial was to explore whether yoga would positively affect balance and flexibility and whether yoga could be used as a primary form of home exercise for female college students to help them stay healthy during the COVID-19 pandemic.

[Location of experiment]

Yoga classroom of Chengdu Normal University.

[Participant Requirements]

1. Female university students in school.

2. Have not practiced yoga before.

3. Will not perform other exercises during the experiment.

4. Take a serious attitude toward attending yoga class on time.

5. Voluntarily participate in this study and sign the informed consent form.

[Exclusion criteria]

1. Having a chronic illness or physical disability.

2. Current or previous yoga experience.

3. Physical condition unsuitable for yoga.

4. Physical conditions unsuitable for sports training.

5. Hypertension, hypoglycemia, cerebellar disease, joint disease or eye disease.

[Experiment time]

From April 1, 2020, to July 31, 2020, every Wednesday and every Saturday evening from 18:00-19:10.

[Participants' Compensation]

Participants who participate in the yoga practice on time and complete all the experimental tasks will earn 200 RMB each class, which will be distributed through Alipay after the study is completed.

-- Enrollment Method

If you meet the above criteria and are willing to participate in this research project, please contact 13438917548 (same mobile and WeChat number).

Question: How do you explain the absence of drop outs?

Answer: Because we paid participants who persisted in completing all tasks in this study 200 RMB, which is relatively high compared to other experiments of the same type in the Wenjiang area of Chengdu, there were no dropouts.

Question: How did you perform the sample size calculation? Was the sample a convenience one? Please specify it.

Answer: We assumed a sample size ratio of 1:1 for the yoga and control groups, a test level of α=0.05 (two-sided), and a test efficacy of 1-β=0.9. SAS 9.4 software was used to calculate the sample size required for this study, which was a total of 40 participants, 20 in the yoga group and 20 in the control group.

Question: Line 164 and everywhere inside the text: no capital letters after the colon.

Answer: We have corrected this error.

Question: Lines 194-195: please reformulate the sentence.

Answer: Thank you for your suggestion. I have rephrased the sentence as you suggested. The yoga group practiced yoga under the guidance of a yoga instructor for 70 minutes twice a week for 16 weeks.

Question: It is not well expressed what treatment underwent the control group. This is mandatory in a randomized controlled trial. Please clearly explain it.

Answer: Thank you for bringing this to our attention. I have added the control group intervention to the manuscript. The control group watched 10 minutes of yoga videos each week for 16 weeks. These yoga videos included Hatha, Aerial, Ashtanga, Iyengar, Flow, Yin and Dance yoga. 

3. Question: DISCUSSION AND CONCLUSIONS

 You should revise the structure of this sections. Conclusions are too long, and a substantial part of them could be moved to the discussion which seems poor and needs to be better contextualised. Moreover, is not clear if You aimed at improving psychological health, which largely mentioned in the introduction, or at improving the balance and flexibility tout court. To better describe the context in which you did this research, highlighting the peculiarities of Covid-19 pandemic and female teenagers psychophysical profile, I suggest the following references: Farì, G., Di Paolo, S., Ungaro, D., Luperto, G., Farì, E., & Latino, F. (2021). The impact of covid-19 on sport and daily activities in an Italian cohort of football school children. International Journal of Athletic Therapy and Training, 26(5), 274-278. doi:10.1123/ijatt.2020-0066 Notarnicola, A., Farì, G., Maccagnano, G., Riondino, A., Covelli, I., Bianchi, F. P.,. .. Moretti, B. (2019). Teenagers’ perceptions of their scoliotic curves. an observational study of comparison between sports people and non-sports people. Muscles, Ligaments and Tendons Journal, 9(2), 225-235. doi:10.32098/mltj.02.2019.11 de Sire, A., Demeco, A., Marotta, N., Spanò, R., Curci, C., Farì, G.,. ... Ammendolia, A. (2022). Neuromuscular impairment of knee stabilizer muscles in a COVID-19 cluster of female volleyball players: Which role for rehabilitation in the post-COVID-19 return-to-play? Applied Sciences (Switzerland), 12(2) doi:10.3390/app12020557

3. Answer: Thank you for your suggestions, these are very good and helpful in improving the manuscript. We have revised the Discussion and Conclusion sections of the manuscript as you suggested. The Discussion has been rewritten, and the Conclusion has been streamlined to more clearly describe the background and purpose of this study. The references you suggested are well written and very professional, and we have cited them. Please review the Discussion and Conclusion sections of this revised manuscript. Thank you very much.

Reviewer #2:

1. Question: KEYWORDS

In order to increase the visibility of the article, do not use keywords already present in the title.

1. Answer: Thank you for your suggestion, which is very well proposed. I have revised the keywords of the manuscript as you suggested; please review the keywords. Keywords: Public health emergencies; health threats; women's health; home exercise

2. Question: INTRODUCTION

The backround is well described. The purpose of the study is clearly stated. "h are facing major threats.". You need a reference. I suggest inserting the following reference: - de Sire A, Marotta N, Agostini F, Drago Ferrante V, Demeco A, Ferrillo M, Inzitari MT, Pellegrino R, Russo I, Ozyemisci Taskiran O, Bernetti A, Ammendolia A. A Telerehabilitation Approach to Chronic Facial Paralysis in the COVID -19 Pandemic Scenario: What Role for Electromyography Assessment? J Pers Med. 2022 Mar 19;12(3):497. doi:10.3390/jpm12030497. PMID: 35330496; PMC ID: PMC8949994

2. Answer: Thank you for your advice. I have followed your suggestion and inserted the reference you recommended for this sentence.

3. Question: METHODS

The inclusion and exclusion criteria have been adequately described. The measurements are in accordance with the aim of the study. Statistical analysis was adequately described and performed. "57 participants who came to the yoga consultation 124 were selected, and 40 participants were enrolled in the study....of 20 women after screening (Figure 1)......articipants (n = 40). The sociodemograph...(Table 1; p > 0.05). ". Move this information, and also figure 1, to the results section. In the methods section, there must be no references to the number of patients, but only inclusion and exclusion criteria.

3. Answer: Thank you for your very professional advice. I have moved the content you specified from the methods section to the results section as you suggested. Please review the revised sections.

Thank you for your very helpful comments and advice. If the manuscript needs additional revision, please do not hesitate to contact us. Thank you very much!

Kind regards,

Xue Luo and Xu Huang

---

## [Editor Report · Decision Letter 1]

13 Feb 2023

The effects of a yoga intervention on balance and flexibility in female college students during COVID-19: A randomized controlled trial

PONE-D-22-33655R1

Dear Dr. Luo,

We’re pleased to inform you that your manuscript has been judged scientifically suitable for publication and will be formally accepted for publication once it meets all outstanding technical requirements.

The authors adequately revised their paper.

The manuscript is acceptable in the present form.

Kind regards,

Alessandro de Sire, M.D.

Academic Editor

PLOS ONE

---

## [Editor Report · Acceptance letter]

13 Mar 2023

PONE-D-22-33655R1 

The effects of a yoga intervention on balance and flexibility in female college students during COVID-19: A randomized controlled trial 

Dear Dr. Luo:

I'm pleased to inform you that your manuscript has been deemed suitable for publication in PLOS ONE. Congratulations! Your manuscript is now with our production department. 

Kind regards, 

on behalf of

Prof. Alessandro de Sire 

Academic Editor

PLOS ONE